# Gene Regulation during Carapacial Ridge Development of *Mauremys reevesii*: The Development of Carapacial Ridge, Ribs and Scutes

**DOI:** 10.3390/genes13091676

**Published:** 2022-09-19

**Authors:** Jiayu Yang, Yingying Xia, Shaohu Li, Tingting Chen, Jilong Zhang, Zhiyuan Weng, Huiwei Zheng, Minxuan Jin, Chuanhe Bao, Shiping Su, Yangyang Liang, Jun Zhang

**Affiliations:** 1College of Animal Science and Technology, Anhui Agricultural University, Hefei 230036, China; 2Fisheries Research Institute, Anhui Academy of Agricultural Sciences, Hefei 230041, China

**Keywords:** *Mauremys reevesii*, carapace ridge, turtle, Wnt signaling pathway, Hedgehog signaling pathway

## Abstract

The unique topological structure of a turtle shell, including the special ribs–scapula relationship, is an evolutionarily novelty of amniotes. The carapacial ridge is a key embryonic tissue for inducing turtle carapace morphologenesis. However, the gene expression profiles and molecular regulatory mechanisms that occur during carapacial ridge development, including the regulation mechanism of rib axis arrest, the development mechanism of the carapacial ridge, and the differentiation between soft-shell turtles and hard-shell turtles, are not fully understood. In this study, we obtained genome-wide gene expression profiles during the carapacial ridge development of *Mauremys reevesii* using RNA-sequencing by using carapacial ridge tissues from stage 14, 15 and 16 turtle embryos. In addition, a differentially expressed genes (DEGs) analysis and a gene set enrichment analysis (GSEA) of three comparison groups were performed. Furthermore, a Kyoto Encyclopedia of Genes and Genomes (KEGG) pathway analysis was used to analyze the pathway enrichment of the differentially expressed genes of the three comparative groups. The result displayed that the Wnt signaling pathway was substantially enriched in the CrTK14 vs. the CrTK15 comparison group, while the Hedgehog signaling pathway was significantly enriched in the CrTK15 vs. the CrTK16 group. Moreover, the regulatory network of the Wnt signaling pathway showed that Wnt signaling pathways might interact with *Fgfs*, *Bmps*, and *Shh* to form a regulatory network to regulate the carapacial ridge development. Next, WGCNA was used to cluster and analyze the expression genes during the carapacial ridge development of *M. reevesii* and *P. sinensis*. Further, a KEGG functional enrichment analysis of the carapacial ridge correlation gene modules was performed. Interesting, these results indicated that the Wnt signaling pathway and the MAPK signaling pathway were significantly enriched in the gene modules that were highly correlated with the stage 14 and stage 15 carapacial ridge samples of the two species. The Hedgehog signaling pathway was significantly enriched in the modules that were strongly correlated with the stage 16 carapacial ridge samples of *M. reevesii*, however, the PI3K-Akt signaling and the TGF-β signaling pathways were significantly enriched in the modules that were strongly correlated with the stage 16 carapacial ridge samples of *P. sinensis*. Furthermore, we found that those modules that were strongly correlated with the stage 14 carapacial ridge samples of *M. reevesii* and *P. sinensis* contained *Wnts* and *Lef1*. While the navajo white 3 module which was strongly correlated with the stage 16 carapacial ridge samples of *M. reevesii* contained *Shh* and *Ptchs*. The dark green module strongly correlated with the stage 16 carapacial ridge samples of *P. sinensis which* contained *Col1a1*, *Col1a2*, and *Itga8*. Consequently, this study systematically revealed the signaling pathways and genes that regulate the carapacial ridge development of *M. reevesii* and *P. sinensis*, which provides new insights for revealing the molecular mechanism that is underlying the turtle’s body structure.

## 1. Introduction

Turtles are living, ancient reptiles, and the lineage of modern turtles split from archosaurs (birds and crocodiles) about 220 to 260 million years ago [1]. The turtle shell has a unique topological structure, which is an extraordinary evolutionary novelty among extant, armored, living tetrapods [2]. It is considered as an excellent model for the study of evolutionary and developmental biology [3,4]. The turtle shell is made up of a carapace that consists of severely curved dorsal vertebrae, ribs, bone plates, and a plastron that protects the abdomen at the bottom [5]. In turtles, their ribs grow laterally in a superficial layer of the dorsal trunk to encapsulate the scapula and pelvis, which is markedly distinct from that of other amniotic animals [6,7,8]. In particular, their ribs never grow into the body wall, and instead, they are arrested within the primaxial dermis, forming a folding structure at the hinge between the body wall and the axis [7,9]. Abundant fossil evidence of stem turtles has showed that the existing body structure of turtles evolved gradually, in multiple steps [10]. The evolution of the “*Eunotosauru-Pappochelys-Eorhynchochelys-Odontochelys-Proganochelys*” was characterized by tooth degeneration, rib broadening, spine reduction, and the progressive integrity of the ventral shell and the carapace [11,12,13]. Modern turtles consist of two branches, the Pleurodira and Cryptodira [14]. Soft-shell turtles are one of the sister groups of Cryptodira, while hard-shell turtles contain both branches within their lineage [14]. Both soft-shell turtles and hard-shell turtles have a turtle-specific shell topological structure [9]. The difference is that the carapace of soft-shell turtles is covered with leathery skin, while the carapace of a hard-shell turtle is covered with keratose scutes, which are derived from the scales of the hard-shell and local epithelial thickenings [15,16]. The scutes of hard-shell turtles were thought to be similar to feathers and were considered as modified scales [1]. From an evolutionary perspective, the scutes first appeared in Proganochelys at least 210 million years ago, and they were regarded as a distinct evolutionary feature that has been preserved in extant hard-shell turtles [17]. The extant Trionychidae and Carettochelyidae were considered to be the most unique derived group, having lost their scutes during their evolution, and these have since been replaced by a thick leathery epidermis [18].

The carapace morphogenesis is induced by a ridge along the dorsal flank of the stage 14 embryos, called the carapacial ridge (CR), which is a unique tissue of turtle embryos [19,20]. The CR is constructed by mesenchymal cells which are encased under an overlying layer of thickened epithelial cells; Burke hypothesized that the carapace development was also regulated by the CR’s structure [20]. Moreover, evidence suggests that the CR may be a molecular signaling center during the turtle embryo’s development, which is responsible for the rib arrangement and the scutes’ development [7,16,21]. Kuraku performed a microbead-based differential cDNA screening, and they found that specificity protein-5 (SP5), cellular retinoic acid-binding protein-I (CRABP-1), lymphoid enhancer-binding factor-1 (LEF1), and adenomatous polyposis coli downregulated 1 (APCDD1) were specifically expressed in the CR and the adjacent lateral body wall of stage 14 embryos of *Pelodiscus sinensis* [6]. The electroporation of the dominant-negative form of *Lef1* resulted in the loss of the CR and a partial disturbance of the proximal ribs, suggesting that *Lef1* may be essential for CR growth and rib alignment [7]. Wang identified that *Wnt5a* (encoding Wnt family member 5A) was the only Wnt gene that was expressed in the CR region of a soft-shell turtle, and Pascual-Anaya identified the same turtle-specific *Wnt5a* expression in hard-shell turtles [22,23]. Studies on the CR of Chinese soft-shell turtles revealed that *Wnt5a* regulates the CR development through the Wnt5a/JUN *N*-terminal Kinase (JNK) pathway [24]. In addition, an in situ hybridization showed that *Fgf8* and *Fgf10* (encoding Fibroblast growth factor 8 and 10) are expressed in the distal tip of the ribs and the mesenchyme beneath the CR ectoderm, respectively, possibly forming a feedback loop [25]. Moustakas and colleagues reported that *Grem* (encoding Gremlin) was expressed in the carapace, and *Pax1* (encoding paired box 1) and *Shh* (encoding Sonic hedgehog) were expressed in the CR. In their later studies, essential genes *Bmp2*, *Bmp4* (encoding Bone morphogenetic protein 2, 4), *Shh*, and *Msx2* (encoding msh homeobox 2) were found to be expressed along the CR, forming overlapping chains in the tissue sections of stage 16 embryos, which were affected by FGF signaling [16,21]. However, previous research hypotheses have not been fully confirmed. Further studies are still needed to elucidate the gene expression and molecular regulatory network that is underlying CR development, including the mechanisms of rib axis arrest, CR formation, and the differentiation between soft-shell turtles and hard-shell turtles.

The Chinese pond turtle (*M. reevesii*) is an excellent model for investigating the CR developmental of hard-shell turtles and further revealing the carapace morphogenesis mechanism of Chelonians. In this study, the RNA sequencing of CR tissues from stage 14 to stage 16 embryos was performed using an Illumina Hiseq 4000 platform to clarify the genetic basis during the CR development of *M. reevesii*. In addition, a differentially expressed genes (DEGs) and a gene set enrichment analysis (GSEA) of three comparison groups were performed. Furthermore, a Kyoto Encyclopedia of Genes and Genomes (KEGG) pathway enrichment analysis of the DEGs of three comparative groups were performed to revealed the dynamic changes of the signaling pathways that are involved in CR development. Moreover, the regulatory networks of the Wnt signaling pathway during CR development were constructed. Next, the WGCNA method was used to cluster and analyze the expression genes during the CR development of *M. reevesii* and *P. sinensis*. Further, a KEGG functional enrichment analysis was performed on the gene modules with high degree of correlation degree with the CR. In this study, the dynamic changes of the signaling pathways and genes that are involved in CR development were compared between hard-shell turtles and soft-shell turtles, providing a new perspective for revealing turtle carapace morphology.

## 2. Materials and Methods

### 2.1. Animals and Sample Collection

Fertilized eggs of *M. reevesii* were cultured in captivity, which were collected from an aquafarm in Anhui, China. The turtle eggs were incubated in vermiculite with the fertilization spot of the eggs facing upwards, with a humidity from 10 to 20% at 30 ± 2 °C. The embryonic developmental stage of *M. reevesii* was determined according to Yntema et al. [26] and Tokita and Kuratani [27]. We collected carapacial ridge tissues that included the structure of the ectoderm mesenchyme from the flank from the stage 14 (CrTK14), stage 15 (CrTK15), and stage 16 (CrTK16) turtle embryos under a stereoscopic microscope using sterile forceps. From 25 to 30 embryos were used as a mixed sample. All samples were stored at −80 °C. All experiments in this research were performed following the permitted guidelines which were established by Anhui Agricultural University, and the experimental protocols which were approved by the Animal Care and Use Committee of Anhui Agricultural University.

### 2.2. RNA Extraction, Library Construction and Sequencing, Data Quality Control and Basic Annotation

The total RNA was extracted from the CR tissues samples using a Trizol reagent kit (Invitrogen, Carlsbad, CA, USA) according to the manufacturer’s specifications [28]. The RNA quality was assessed and checked using an Agilent 2100 Bioanalyzer (Agilent Technologies, Santa Clara, CA, USA) and RNase-free agarose gel electrophoresis. Then, Oligo (dT) beads were used to enrich the mRNA, and the rRNA was eliminated using a Ribo-ZeroTM Magnetic Kit (Epicentre, Madison, WI, USA). After the mRNA’s reverse transcription into cDNA, the cDNA was purified using a QiaQuick PCR extraction kit (Qiagen, Venlo, Netherlands). Gene Denovo Biotechnology Co. (Guangzhou, China) sequenced the cDNA using an Illumina HiSeqTM 4000 sequencer [28].

The raw reads that were obtained from the sequencing machine included high-quality, clean reads and low-quality reads, and also, substandard reads would have affected the following assembly and analysis. Thus, the reads were filtered using fastp [29] to obtain the high-quality, clean reads. The reads that contained adapters or those containing more than 10% unknown nucleotides (N), or low-quality reads containing more than 50% low quality (Q ≤ 20) bases were filtered out. After filtering, the high-quality reads accounted for over 99% of the sample, while the low-quality reads accounted for about 0.3%, and no reads contained more than 10% unknown nucleotides (N) in this sample [30]. Then, the transcriptome completeness was assessed used the Benchmarking Universal Single-Copy Orthologs (BUSCO) database (www.orthodb.org (accessed on 1 September 2019)).

The BLASTx program was used to align the unigenes with the proteins in the order of the NCBI non-redundant protein (Nr) database, the Swiss-Prot protein database, the Kyoto Encyclopedia of Genes and Genomes (KEGG) database, and the Clusters of Orthologous Groups of protein (COGs)/EuKaryotic Orthologous Groups (KOG) database. These four databases were searched using an E-value threshold that was under 1 × 10^−5^. According to the best alignment results, protein functional annotations were obtained [31].

### 2.3. Differentially Expressed Genes

The gene abundances were calculated and normalized to reads per kilobase of transcript per million mapped reads (RPKM) before an RNA differential expression analysis was performed. After eliminating the influence of different gene lengths and the amount of sequencing data in the calculation of the gene expression, the calculated gene expression could be used directly to compare the differences in gene expression among the samples. The read counts from the previous step were analyzed using DESeq2 [32] of an RNA differential expression. We screened the significantly differentially expressed genes (DEGs) between the samples with a false discovery rate of (FDR) ≤ 0.05 and a |log2(fold change (FC))| ≥ 1.

### 2.4. qRT-PCR

A quantitative real-time reverse transcription PCR (qRT-PCR) was performed to verify the gene expression trends. The total RNAs were extracted from the stage 14, 15, and 16 turtle CR tissue samples. Then, the RNA was reverse-transcribed into cDNA, which was used as the template for a quantitative real-time PCR examination. The primer sequences for the genes are shown in Table 1. The *Gapdh* gene (encoding glyceraldehyde-3-phosphate dehydrogenase) was used as the internal reference gene in *M. reevesii* [33]. Each experiment was repeated three times.

### 2.5. Gene Set Enrichment Analysis (GSEA)

A GSEA was performed using the software GSEA [34] and MsigDB [35], which were used to search whether the set of genes expressed significant differences between the two comparison groups. The gene expression matrix and rank genes were subjected to the Signal-to-Noise normalization method, using the default parameters, to calculate the enrichment scores and *p*-values.

### 2.6. Pathway Enrichment Analysis

A pathway enrichment analysis was performed according to the KEGG pathway analysis which was based on the KEGG public databases [36]. The KEGG annotation was based on the whole genome background. The enrichment results were used to find the significantly enriched metabolic pathways or signal transduction pathways involving DEGs and compare them with the whole genome background. The formula that was used to calculate the *p*-value was as follows:$$P=1-\overset{m-1}{\underset{i=0}{\sum }}\frac{\left(\begin{array}{c} M\\ i \end{array}\right)\left(\begin{array}{c} N-M\\ n-i \end{array}\right)}{\left(\begin{array}{c} N\\ n \end{array}\right)}$$

In this formula, the numbers of all the genes and DEGs in the KEGG annotation are N and n, respectively; the numbers of all the genes and DEGs that were annotated to be involved with specific pathways are M and m, respectively. After taking the correction of FDR ≤ 0.05 as a threshold to correct them, the q-values of the enriched pathways in DEGs were defined [37].

### 2.7. Short Time-Series Expression Miner (STEM) Analysis

We performed a STEM analysis to cluster the DEGs with similar expression patterns. By normalizing the expression pattern at the stage 14 expression level of each sample to 0, a significant clustering was identified at *p* ≤ 0.05 [38].

### 2.8. Constructing a Network of Key Genes

The online tool String v10 (string-db.org) was used to analyze the protein interactions that were associated with the Wnt signaling pathway [39], based on the criterion of a combined score > 900; then the network was constructed and exhibited using Cytoscape software [40].

### 2.9. Weighted Gene Co-Expression Network Analysis (WGCNA)

A WGCNA was used to identify the features among genes in different samples. The WGCNA package that is in R language was used to construct the co-expression network. The gene expression values were imported to the WGCNA to identify the expression modules after filtering the genes [41]. The correlation coefficients between the samples and sample traits were calculated using module Eigengenes. A GO and KEGG enrichment analysis were performed to identify the biological functions of the gene modules.

## 3. Results

### 3.1. De Novo Transcriptome Assembly and Annotation

To analyze the genetic mechanism of the CR development of *M. reevesii* on a genome-wide scale, we performed RNA sequencing of the CR tissues from stage 14 to stage 16 turtle embryos. Three samples from each stage were sequenced in duplicate. A total of 351,359,694 raw reads were generated, thus leaving 350,061,012 clean reads after filtering, therefore accounting for 99.63% (Table 2). A total of 123,036 unigenes were assembled since no reference genome of the CR tissues of *M. reevesii* has been assembled. The N50 analysis showed that the average unigene length among all the samples was 904 bp, the N50 number was 14,326, and the N50 length was 1946 bp. The total amount of unigenes comprising the 978 groups were searched using BUSCO; the number of complete BUSCOs was 873 (89.3%), including 863 complete single copies and 10 complete duplicates. The number of fragmented and missing BUSCOs were 72 and 33, respectively (Appendix A). Next, 123,036 unigenes were annotated using data in the Nr, KEGG, COG, and SwissProt databases. The functional annotation results showed that there are 39,906 unigenes in the databases which consisted of 39,703 unigenes in Nr, 27,951 unigenes in KEGG, 14,791 unigenes in COG, and 19,221 unigenes in SwissProt. To analyze the corresponding homologous sequences and identify the similar species that the homologous sequence belongs to, annotations in the Nr database were used. According to Nr annotation results, 15,907 (40.1%) unigenes were homologous to the *Chrysemys picta bellii* sequences and 13,680 (34.5%) unigenes were homologous to the *Chelonia mydas* sequences, while only 1736 (4.4%) unigenes were homologous to the *P. sinensis* sequences (Figure 1).

### 3.2. Analysis of Differentially Expressed Genes

To screen for the differentially expressed genes (DEGs) during CR development, a DEGs analysis of three comparison groups, including CrTK14 vs. CrTK15, CrTK14 vs. CrTK16, and CrTK15 vs. CrTK16, was performed. The genes were considered to be DEGs when the FDR < 0.05 and the |log2FC| ≥ 1. The results showed that there were 324 upregulated DEGs and 796 downregulated DEGs in the CrTK14 vs. CrTK15 group, 198 upregulated DEGs and 781 downregulated DEGs in the CrTK15 vs. CrTK16 group, and 855 upregulated DEGs and 2686 downregulated DEGs in the CrTK14 vs. CrTK16 group (Figure 2A). The volcano plots of three comparison groups are displayed in Figure 2B–D. According to the *p*-values of the total number of DEGs, Fibroblast growth factor (*Fgf*), Bone morphogenetic protein (*Bmp*), Sry-like high-mobility group box (*Sox*), and Wingless-type (*Wnt*) ranked at the top of the list, based on the *p*-values (Appendix A). Then, heat maps of 20 representative genes were drawn (Figure 2E) for *Bmp5*, *Bmp6*, *Fgf8*, *Fgfr1*, *Msx2*, *Msxc*, *Pax1*, *Pax3*, *Sox8*, *Sox9*, *Wnt11* and *Wnt16* which were downregulated during CR development, and *Bmp3*, *Bmp7*, *Dkk1*, *Fgf10*, *Lef1*, *Shh*, *Sp5*, and *Wnt5a* which were upregulated during CR development.

Next, the DEGs were analyzed by a gene set enrichment analysis (GSEA), which could identify the functional individual genes with subtle changes in their expression. The genes with similar expression patterns were classified into a gene set and each comparison contained two kinds of gene sets, including the genes that were upregulated and downregulated in that comparison group, respectively. All gene sets from the three comparison groups were analyzed by a gene ontology (GO) enrichment analysis. A larger absolute value of the normalized enrichment score (NES) was considered to indicated a more significant enrichment. As shown in Figure 3, the upregulated and downregulated genes sets that were expressed are displayed as red and green lines, respectively. “Cardiac chamber development (GO:0003205)” was the most significantly enriched term in the CrTK14 vs. CrTK15 group, “endothelial cell differentiation (GO:0045446)” was the most significantly enriched term in the CrTK15 vs. CrTK16 group, and the “obsolete contractile fiber part (GO:0044449)” was a strongly enriched term in the CrTK14 vs. CrTK16 group (Figure 3A–C). Notably, “regulation of epithelial cell proliferation (GO:0050678)” and “epithelial cell proliferation (GO:0050673)” were the significant enriched terms in the CrTK14 vs. CrTK15 group, wherein the tNES values were 1.57 and 1.48, respectively (Figure 3D–E). Additionally, the NES values of the “smoothened signaling pathway (GO:0007224)” was 2.33 in the CrTK15 vs. CrTK16 group (Figure 3F). In addition, the GO enrichment analysis of the significantly downregulated gene sets displayed that “cardiac chamber development (GO:0003206)” and “kidney epithelium development (GO:0072073)” were significantly the negatively enriched terms in the CrTK14 vs. CrTK15 group (Figure 3G–H). Moreover, the “striated muscle cell development (GO:0055002)” and “obsolete contractile fiber part (GO:0044449)” were the significantly negatively enriched terms in the CrTK15 vs. CrTK16 group (Figure 3I–J and Appendix A).

### 3.3. KEGG Pathway Enrichment Analysis

To further reveal the biological functions of the DEGs during CR development, a KEGG enrichment analysis of the DEGs was performed, for which, the q-value was calculated after the *p*-value through an FDR Correction (Figure 4). Interesting, it was found that the PI3K-Akt signaling pathway, the cGMP-PKG signaling pathway, the cAMP signaling pathway and the Calcium signaling pathway ranked in the top 10 of the signaling pathway enrichment diagram for the three comparative groups (Figure 4A–C). Remarkably, the signaling pathways that were ranked in the top 15 of the signaling pathway enrichment test were different between the CrTK14 vs. CrTK15 group and the CrTK15 vs. CrTK16 group studies. Such as, the Wnt signaling pathway was ranked in top two in the CrTK14 vs. CrTK15 group, while it was ranked in in the top five in the CrTK15 vs. CrTK16 group. The Hedgehog signaling pathway was not ranked in top 15 in the CrTK14 vs. CrTK15 group, while the Hedgehog signaling pathway was ranked as the in top pathway in the CrTK15 vs. CrTK16 group (Figure 4). Furthermore, heat maps of the Wnt signaling pathway and Hedgehog signaling pathway were drawn during the CR development (Figure 5). As shown in Figure 5A, after stage 14, Frizzled (encoded by *Fzd10*), the receptor of the Wnt signaling pathway, was upregulated. DKK (encoded by *Dkk4*), the antagonist of the Wnt signaling pathway, was also upregulated after stage 14. The genes in the canonical pathway, such as those encoding upstream Wnt ligands (*Wnt9* and *Wnt10*) and the downstream gene (*Lef1*), were upregulated after stage 15. While the *Wnt11* gene, encoding the upstream ligand of the non-canonical Wnt signaling pathway, was downregulated from stage 14 to stage 16, whereas *Wnt5a* was upregulated. As shown in Figure 5B, the receptor Patched (*Ptch1*, *Ptch2*) and *Shh* in the Hedgehog signaling pathway were upregulated from stage 15 to stage 16. These results indicate that the Wnt signaling pathway was significantly enriched in early stage of the CR development, while the Hedgehog signaling pathway was significantly enriched in late stage of the CR development.

### 3.4. DEGs Expression Trend Analysis

To analyze the expression trends of the DEGs during CR development, 3915 DEGs were analyzed using STEM software, and the results showed that there were eight gene expression profiles (Figure 6). As shown in Figure 6, 1461 DEGs belonged to Profile 0, and their expression level was downregulated from stage 14 to 16. Four hundred and forty-one DEGs belonged to Profile 1 and their expression was downregulated from stage 14 to 15, then no significant change was observed from stage 15 to 16. Forty-five DEGs belonged to Profile 2 and their expression levels were downregulated from stage 14 to 15, then their expressed was upregulated from stage 15 to 16. Nine hundred and fifty-six DEGs belong to Profile 3 and their expression level did not change significantly from stage 14 to 15, then they were downregulated from stage 15 to 16. One hundred and twenty DEGs belonged to Profile 4 and their expression level was slightly downregulated from stage 14 to 15, then they were upregulated from stage 15 to 16. Seventy-one DEGs belonged to Profile 5, whose expression level was upregulated from stage 14 to 15 and downregulated from stage 15 to 16. Three hundred and forty-three DEGs belonged to Profile 6 and their expression level was upregulated from stage 14 to 15, but there was no significant change from stage 15 to 16. Four hundred and seventy-eight DEGs belonged to Profile 7 and their expression level was continuously upregulated from stage 14 to 16. In addition, *Wnt11* and *Pax1* belonged to Profile 0, and their expression levels were downregulated from stage 14 to 16. *Fgf8* belonged to Profile 3 and its expression level was significantly downregulated from stage 15 to 16. *Wnt5a*, *Lef1* and *Sp5* belonged to Profile 6 and their levels were upregulation from stage 14. *Dkk1* and *Dkk4* belonged to Profile 4 and Profile 7, while *Shh* and *Ptch1* belonged to Profile 4, and their expression levels were upregulated from stage 15 to stage 16 (see Appendix A). These results showed that the Wnt signaling pathway related genes and Fgf genes were mostly upregulated from stage 14, while the Hedgehog signaling pathway related genes and Wnt antagonist genes were upregulated from stage 15.

### 3.5. Real-Time RT-PCR Validation

Furthermore, a qRT-PCR was used to quantitatively analyze the expression of *Fgf8*, *Bmp5*, *Msx2*, *Wnt5a*, *Wnt11*, *Sp5*, *Pax1*, and *Lef1* during the CR development of *M. reevesii*. The results showed that the gene expression trends were almost consistent with those in the RNA-seq data (Figure 7).

### 3.6. Interaction Network of Wnt Signaling Pathway

In order to reveal the regulatory network of the Wnt signaling pathway during CR development, the protein–protein interaction of the Wnt signaling pathway that is related genes was analyzed using the String database (Appendix A). Relationships with scores of more than 900 were selected and mapped using the Cytoscape software. The network contains 134 genes and 1110 pairs of gene interaction relationships. Moreover, nine genes, including *Wnt5a*, *Wnt3a*, *Wnt11*, *Lef1*, *Bmp2*, *Bmp4*, *Bmp5*, *Shh*, *Fgf8*, and *Fgf1* were placed in the center of the network and highlighted in yellow (Figure 8). Both *Fgf8* and *Fgf10* correlated with *Fgfr2*, *Shh*, and *Bmp4*. *Bmp4* correlated with *Shh* and *Bmp2*. *Wnt8c*, *Wnt3a*, and *Wnt2b* correlated with *Wnt-1*, and *Wnt-1* correlated with *Fgf8*. *Wnt11* correlated with *Fzd7* and *Dvl2*. *Wnt5a* correlated with *Vangl2* and *Dkk1*. The results indicated that the Wnt signaling pathways might interact with *Fgf*, *Bmp*, and *Shh* to form a complex regulatory network that is involved in CR development.

### 3.7. Weighted Co-Expression Network Analysis (WGCNA) of CR

A WGCNA was performed to cluster and analyze the genes with similar expression patterns during the CR development of *M. reevesii* and *P. sinensis* (Appendix A). The transcriptome data from stage 14 (TK14), stage 15 (TK15) and stage 16 (TK16) CRs of *P. sinensis* (SRA: PRJNA742492) were used for the analysis. The results of the WGCNA analysis showed that the CR transcriptome genes of *M. reevesii* were divided into 22 gene modules (Figure 9A), and the CR transcriptome genes of *P. sinensis* were divided into 20 gene modules (Figure 10A). Heat maps were used to show the correlations between the gene modules, and a Pearson’s correlation coefficient indicated the degree of correlation between the two modules (Figure 9B and Figure 10B). The correlation between the modules and samples was shown through the heat map, with red to green representing the correlation from high to low, respectively. The results showed that the cyan module was highly correlated with the stage 14 CR samples of *M. reevesii*, whereas the brown 3 module, the dark magenta module, and the coral 3 modules were highly correlated with the stage 15 CR samples of *M. reevesii*. The navajo white 3 module was significantly correlated with the stage 16 CR samples of *M. reevesii* (Figure 9C). The module and sample correlation heat map of *P. sinensis* showed that the black module was significantly correlated with the stage 14 CR samples. The pink and coral 1 modules were significantly correlated with the stage 15 CR samples and dark green module was highly correlated with the stage 16 CR samples (Figure 10C).

In addition, a KEGG functional enrichment analysis of the CR correlation modules of *M. reevesii* and *P. sinensis* was performed, respectively. The top 20 KEGG enrichment pathways of the cyan module, which were associated with the stage 14 CR samples of *M. reevesii*, contained the MAPK signaling pathway and the TGF-β signaling pathway (Figure 11A). The top 20 KEGG enrichment pathways of the black module, which were associated with the stage 14 CR samples of *P. sinensis*, contained the Wnt signaling pathway and the MAPK signaling pathway (Figure 11E). The top 20 KEGG enrichment pathways of the brown 3 module, which were associated with the stage 15 CR samples of *M. reevesii*, contained the Wnt signaling pathway (Figure 11B). The top 20 KEGG enrichment pathways of the navajo white 3 module, which were associated with the stage 16 CR samples of *M. reevesii*, contained the Hedgehog signaling pathway, while the top 20 KEGG enrichment pathways of the dark green module, which were associated with the stage 16 CR samples of *P. sinensis* contained the PI3K-Akt signaling and the TGF-β signaling pathways (Figure 11D,H). These results indicated that the Wnt signaling pathway and/or the MAPK signaling pathway were found among the top 20 enrichment pathways of all of the modules which were highly correlated with the stage 14 and stage 15 CR samples of the two species. However, the Hedgehog signaling pathway was significantly enriched in the modules that were strongly correlated with the stage 16 CR samples of *M. reevesii*, while the PI3K-Akt signaling and the TGF-β signaling pathways were significantly enriched in the modules that were strongly correlated with the stage 16 samples of *P. sinensis*.

### 3.8. Comparison and Analysis of Gene Modules Correlated with CR of M. reevesii and P. sinensis

Furthermore, the comparison and analysis of eight modules that were strongly correlated with the CR samples of *M. reevesii* and *P. sinensis* showed that the cyan module that was strongly correlated with the stage 14 CR samples of *M. reevesii*, and the black module that was strongly correlated with the stage 14 CR of *P. sinensis* contained Wnt signaling pathway-related genes such as *Wnts* and *Lef1*. In addition, the dark magenta module, which was strongly correlated with the stage 15 CR samples of *M. reevesii*, contained *Jnk* and *Fgfr1*. Meanwhile, the pink module, which was correlated with the stage 15 CR samples of *P. sinensis*, also contained *Jnk* and *Fgfr2*. Interesting, the differences were also found in those modules with a strong correlation with the stage 15 CR samples, such as the coral 3 module of *M. reevesii* which contained *Grem1*, and the coral 1 module of *P. sinensis* which contained *Col1a1* (collagen, type I, and α) and *Itga7* genes. Moreover, the navajo white 3 module, which was strongly correlated with the stage 16 CR samples of *M. reevesii*, contained *Shh* and *Ptchs* (encoding SHH receptors). While the dark green module which was strongly correlated with the stage 16 CR samples of *P. sinensis* contained *Col1a1*, *Col1a2*, and *Itga8* (see Table 3).

## 4. Discussion

The CR was considered as a key embryonic tissue to induce the development of the turtle’s unique body structure [7,8,19]. In this study, we obtained genomic-wide gene expression profiles during the CR development of *M. reevesii*. We revealed that the Wnt signaling pathway was significantly enriched in early stages of CR development, while the Hedgehog signaling pathway was significantly enriched in late stages of CR development. Furthermore, the interaction network of the Wnt signaling pathway displayed that the Wnt signaling pathway may interact with *Fgf*, *Bmp* and *Shh* to form a complex network that regulates the CR development of *M. reevesii*. Moreover, a WGCNA was used to cluster and analyze the expression genes during the CR development of *M. reevesii* and *P. sinensis*. Further, a KEGG functional enrichment analysis of the CR correlation gene modules was performed. Interestingly, we found that the Wnt signaling pathway and the MAPK signaling pathway were significantly enriched in the modules that were highly correlated with the stage 14 and stage 15 CR samples of the two species. The Hedgehog signaling pathway was significantly enriched in the modules that were strongly correlated with the stage 16 CR samples of *M. reevesii*, while the PI3K-Akt signaling and the TGF-β signaling pathways were significantly enriched in the modules that were strongly correlated with the stage 16 CR samples of *P. sinensis*.

The turtle is a typical example of the evolutionary innovation of amniotic animals, and its carapace has a reverse topological structure [42]. The turtle carapace contains ribs and thoracic vertebrae that are fused with neural bones, the anterior nuchal bone, and the peripheral bones [19]. The ribs extend into the dorsal dermis but do not invade the ventral body wall, the scapulae are located inside the ribs, forming the unique thoracic structure of the turtles [7]. Unlike hard-shell turtles, soft-shell turtles lack scutes and peripheral bones, instead, they have leathery skin and skirt tissue along the edges of the carapace [1]. The CR is a temporary tissue that is peculiar to the turtle embryo, which consists of a bulge of mesenchymal cells that are surrounded by a thickened somite-derived dermis on the flank of the turtle embryo [20]. A lack of a CR can lead to distal rib abnormalities and rib fan-shaped-like structure destruction [7]. The morphology of a turtle embryo resembles that of other tetrapod embryos before they reach stage 13, but a bulge of discontinuous CR tissue appears on the embryo’s flank at stage 14. Then, the CR thickens and extends on the flank side of the carapace, forming the edge of the dorsal and ventral body walls [19]. The turtle embryo stages from 13 to 16 encompass the initiation and maturity of the CR [20]. From stage 14 to stage 15, the scute development initiates with the *Bmp2* expression in paired domains of each scute site. In addition, the first reaction–diffusion system appears in stage 15 to 16, when *the Shh* and *Fgf* signaling regulates the pattern of arrayed scute placodes [16]. In our study, we revealed that Wnt genes are significantly upregulated in the CrTK14 vs. CrTK15 group, while *Shh* and Ptchs are significantly upregulated in the CrTK15 vs. CrTK16 group (Figure 2). The Wnt signaling pathway was considered as a key signaling pathway that is involved in CR development and rib alignment [7,20]. Kuraku reported that the canonical Wnt signaling pathway was activated in the CR ectoderm, which play roles in the carapace formation [6]. *Lef1* interacts with β-catenin to activate the transcription of the Wnt target genes [43,44]. *Sp5* is considered as a direct downstream target of the Wnt signaling pathway [45]. Nagashima and his colleagues used ovo electroporation to partly silence *Lef1* in the CR, which caused a partial arrest of the CR development in *P. sinensis* [7]. In our study, *Lef1* and *S**p5* were both upregulated from stage 14 to stage 15 and their expression level tended to be stable after stage 15 (Figure 5). These results have demonstrated that the canonical Wnt signaling pathway was activated from stage 14 to stage 15 during the CR elongation of the *M. reevesii* embryo. In addition, we detected that *Wnt5a* was upregulated from stage 14 to stage 15 in the CR development of *M. reevesii*, whereas *Wnt11* expression was downregulated in the three comparison groups (Figure 5). The β-catenin-independent Wnt/planar cell polarity (PCP) pathway was considered to regulate the convergent movement of the A-P axis elongation in vertebrates [46]. *Wnt5a* is thought to have multiple functions in vertebrate embryos, and induces the phosphorylation of VANGL2, a core protein, in a dose-dependent manner [47,48]. FZD7 mediates Wnt11 activity for the shape change and migration of cells in convergent extension movements during vertebrate gastrulation [49]. *Wnt5a* regulates the CR development in Chinese soft-shell turtles through the Wnt5a/JNK pathway [24]. Notably, in this study, *Dkk1* was upregulated in the CR development of *M. reevesii* after stage 15 (Figure 5). The *Dkk1* gene encodes the antagonist of the Wnt signaling pathway, which can antagonize both canonical and non-canonical Wnt signaling [50]. Our study indicated that the Wnt signaling pathway was activated mainly from stage 14 to stage 15 in the CR development of *M. reevesii*. In addition, we found that the Hedgehog signaling pathway-related genes were significantly upregulated, and it emerged as the most significantly enriched signaling pathway in the CrTK15 vs. CrTK16 group (Figure 4C). Moustakas et al. demonstrated that Hedgehog signaling was necessary for *Shh* expression in the carapace [16]. They also found that the expression of *Shh* segmented patterns were lost in the CR with inhibition of Bmp and Hedgehog signaling. Moreover, this *Shh*-segmented expression were also absent in the embryos of *P. sinensis* [16]. *Shh* and its receptors genes of the Hedgehog signaling pathway, *Ptch1* and *Ptch2*, were expressed from stage 14 to stage 16, and the dramatic upregulated trend of these genes in the CrTK15 vs. CrTK16 group implies that the Hedgehog signaling pathway was initiated at a later stage of the CR development. In conclusion, this study suggested that the Wnt signaling pathway may be activated before stage 15 of the CR development to regulate the carapace morphogenesis, while the Hedgehog signaling pathway may be activated after stage 16 of the CR development to regulate the scutes’ formation.

In this study, we detected the expression of *Fgf8* and *Fgf10* in the CR of *M. reevesii* (Figure 2E). Moreover, the qRT-PCR results showed that the expression of *Fgf8* was downregulated during CR development. This results was consistent with Pascual-Anaya’s report, that *Fgf8* was expressed at the rib’s tip in *P. sinensis*, within a dynamic and restricted somatic domain that was flanking the CR, which was difficult to detect [23]. In addition, as shown in Figure 8, *Fgf10* and *Fgf8* were correlated with both *Fgfr2* and *Shh*, and *Wnt-1* was correlated with *Fgf8*, while *Wnt2b*, *Wnt8c* and *Wnt3a* were all correlated with *Wnt-1*. *Wnt5a was* correlated with *Dkk1* and *Vangl2*, and *Wnt11 was* correlated with *Fzd7* and *Dvl2* in the network. *Fgf8* was correlated with *Bmp4*, *Bmp4* was correlated with *Shh*, and *Bmp2* was correlated with *Bmp4*. There have been reports that *Fgf10* was expressed in the limb mesenchyme, and it induces *Fgf8* expression, forming an *Fgf10*-*Fgf8* loop during limb induction [51]. *Fgf10* was able to maintain *Shh* and *Fgfr2* expression, and it was the only significantly expressed receptor during the AER formation [52,53]. *Wnt2b*, *Wnt8c*, and *Wnt3a* regulate the FGF8/FGF10 loop in limb development and induce AER production [54]. *Wnt5a* plays a permissive role in FGF signaling to induce the establishment of PCP orientation in the limb mesenchyme [55]. A loss-of-function point mutant of the PCP core gene, *Vangl2*, affected the FGF–BMP interaction balance to change the limb growth in mice [56]. Experiments have disturbed *Shh*, Bmp and Fgf signaling, leading to the destruction of segmental pattern expression and the scutes’ development [16]. Moreover, *Wnt5a* and *Wnt11*, and their antagonist DKK1 (Dickkopf1), have biphasic regulatory effects on the Wnt/β-catenin signaling pathway during cardiac cell development, including the regulation of cell specification and inhibition differentiation [57,58]. In limb polydactyly, *Bmp* ligands can inhibit *Fgf* signals to regulate differentiation, and its activity in limbs is inhibited by *Gremlin1*, which is promoted by *Shh* [59]. Therefore, we suggest that the Wnt signaling pathways might regulate the CR development of *M. reevesii* through interacting with *Fgfs*, which interacts with *Bmps* and *Shh* to form a complex network. Previously, we have reported that FGFs and Wnts co-regulate the CR development in Chinese soft-shell turtles [24], which suggested that the molecular mechanisms of CR development between hard-shell and soft-shell turtles might be conserved.

In this study, a WGCNA was used to cluster and analyze the gene expression patterns during the CR development of hard-shell turtles and soft-shell turtles by using CR transcriptome data of *M. reevesii* and *P. sinensis* (SRA: PRJNA742492). Furthermore, a KEGG pathway enrichment analysis was performed on the gene modules with a high correlation in the CR samples from different stages. We found that the Wnt signaling pathway and the MAPK signaling pathway were enriched to the degree that they were in the top 20 pathways of the modules that were highly correlated with the stage 14 and stage 15 CR samples from both species. The Hedgehog signaling pathway was significantly enriched in the modules that were strongly correlated with the stage 16 CR samples of *M. reevesii*, while the PI3K-Akt signaling and TGF-β signaling pathways were significantly enriched in the modules that were strongly correlated with the stage 16 CR samples of *P. sinensis*. Further, we found that both the cyan module that was strongly correlated with the stage 14 CR samples of *M. reevesii* and black module that was strongly correlated with the stage 14 CR samples of *P. sinensis* contained *Wnt* and *Lef1*. In addition, the dark magenta module, which was strongly correlated with the stage 15 CR samples of *M. reevesii*, contained *Jnk* and *Fgfr1*. Meanwhile, the pink module, which was correlated with stage 15 CR samples of *P. sinensis*, also contained *Jnk* and *Fgfr2*. Interestingly, the differences were also found in those modules with a strong correlation with the stage 15 CR samples, for example, coral 3 contained *Grem1* and Coral 1 contained *Col1a1* (collagen, type I, α) and *Itga7* genes. Moreover, the navajo white 3 module, which was strongly correlated with the stage 16 CR samples of *M. reevesii*, contained *Shh* and *Ptchs* (encoding SHH receptors). While the dark green module which was strongly correlated with the stage 16 CR samples of *P. sinensis* contained *Col1a1*, *Col1a2*, and *Itga8*. Type I collagen, a structural protein in the extracellular matrix, is made up of three chains, including two α1(I) chains and α2(II) chains to form a triple helix, which it is controlled by *Col1a1* and *Col1a2*, respectively [60,61]. The receptor of *Col1as* in the PI3K-Akt signaling pathway was ITGA, and the PI3K-Akt signaling pathway was significantly enriched in the gene modules that were associated with stage 16 CR samples. TGF-β can promote the transcriptional activation of the collagen interstitial cells through the stimulation of fibroblasts that differentiate from mesenchymal cells [62,63,64]. The inhibition of TGF-β signaling negatively regulates collagen homeostasis [65]. The skirt tissue that is derived from the posterior edge of the carapace of Chinese soft-shell turtles contains an abundant amount of type I collagen [66]. *Col1a1*, *Col1a2*, and *Itga8* may regulate collagen synthesis in CR through TGF-β signaling and PI3K-Akt signaling. Therefore, we conclude that the genes regulating the CR development of hard-shell turtles and soft-shell turtles differentiate at a later stage of CR development, that is, the genes that are regulating the scute development in hard-shell turtles are activated at a later stage, while the genes that are regulating the skirt tissue formation of soft-shell turtles are activated at the same stage. In addition, the different expression location of the key genes that are expressed in soft-shell turtles and hard-shell turtles also contribute to this differentiation. However, further experiments are needed to confirm the mechanisms underlying the developmental differences between hard-shell turtles and soft-shell turtles.

In conclusion, we comprehensively analyzed the gene expression profile of the CR development of *M. reevesii*, and we have revealed the dynamic changes of the signaling pathways and the expression trends of key genes during CR development. The Wnt signaling pathway was found to be significantly enriched in the early stage of CR development, from stage 14 to 15, which might be involved in CR development and rib induction. While the Hedgehog signaling pathway was significantly enriched in the late stage of CR development, from 15 to 16, which may be related to scute formation. Moreover, the interaction network of the Wnt signaling pathway showed that Wnts interact with *Fgfs*, which interact with *Bmps* and *Shh* to form a complex network during CR development. Furthermore, the Wnt signaling pathway and the MAPK signaling pathway were so enriched as to be in the top 20 pathways of the modules that were highly correlated with the stage 14 and stage 15 CR samples from the two species. However, the Hedgehog signaling pathway was significantly enriched in the modules that were strongly correlated with the stage 16 CR samples of *M. reevesii*, while the PI3K-Akt signaling and TGF-β signaling pathways were significantly enriched in the modules that were strongly correlated with the stage 16 samples of *P. sinensis*. Interestingly, we found that the cyan module that was strongly correlated with the stage 14 CR samples of *M. reevesii* and the black module that was strongly correlated with the stage 14 CR of *P. sinensis* contained *Wnt* and *Lef1*. In addition, the dark magenta module, which was strongly correlated with the stage 15 CR samples of *M. reevesii*, contained *Jnk* and *Fgfr1*. Meanwhile, the pink module, which was correlated with the stage 15 CR samples of *P. sinensis*, also contained *Jnk* and *Fgfr2*. The differences were also found in those modules with a strong correlation with the stage 15 CR samples, for example, coral 3 contained *Grem1* and Coral 1 contained *Col1a1* (collagen, type I, α) and *Itga7* genes. Moreover, the navajo white 3 module, which was strongly correlated with the stage 16 CR samples of *M. reevesii*, contained *Shh* and *Ptchs* (encoding SHH receptors). While the dark green module which was strongly correlated with stage 16 CR samples of *P. sinensis* contained *Col1a1*, *Col1a2*, and *Itga8*. This study systematically revealed the signaling pathways and genes that regulate the carapacial ridge development of *M. reevesii* and *P. sinensis*, which provides new insights for revealing the molecular mechanism that is underlying the turtle’s body structure.

## Figures and Tables

**Figure 1 genes-13-01676-f001:**
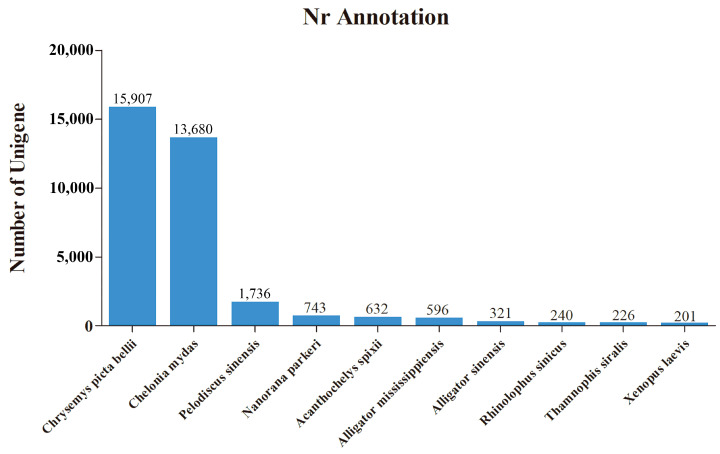
Top 10 homologous sequences of the unigenes from the CR tissues of *M. reevesii* which were annotated in the NCBI non-redundant protein (Nr) database (with an E-value threshold le-5).

**Figure 2 genes-13-01676-f002:**
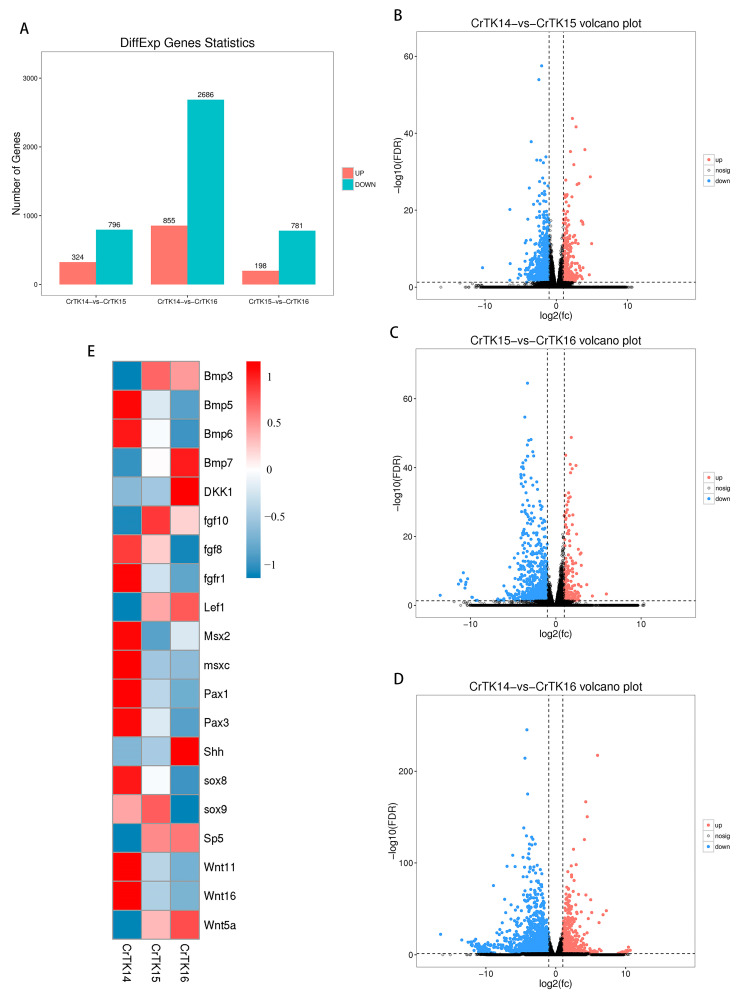
Analysis of differentially express genes (DEGs) during the CR development of *M. reevesii*. (**A**) Histogram of DEGs. Upregulated genes are shown in red. Downregulated genes are shown in blue. (**B**–**D**) Volcano plots of three comparison groups. Red dots indicate upregulated DEGs, blue dots indicate downregulated DEGs, and black dots indicate genes that are not differentially expressed. (**E**) Heat map of 20 DEGs. The DEGs have log-transformed expression values. The values in the corresponding spectrum are 1, 0.5, 0, and −0.5, −1. Blue represents a lower relative expression and red represents a higher relative expression.

**Figure 3 genes-13-01676-f003:**
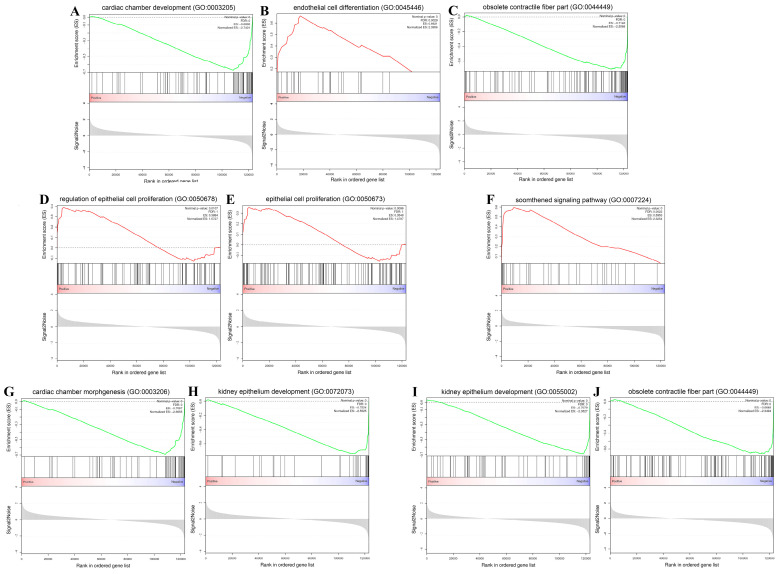
Significant enrichment of gene ontology (GO) plots based on gene set enrichment analysis (GSEA). (**A**–**C**) GO plots of the most significantly enriched gene sets in CrTK14 vs. CrTK15, CrTK15 vs. CrTK16 and CrTK14 vs. CrTK16 groups. (**D**,**E**) “Epithelial cell proliferation” and “regulation of epithelial cell proliferation” were significantly enriched in CrTK14 vs. CrTK15. (**F**) “Smoothened signaling pathway” was significantly enriched in CrTK15 vs. CrTK16. (**G**,**H**) The top 2 negative enrichment in CrTK14 vs. CrTK15. (**I**,**J**) The top 2 negative enrichment in CrTK15 vs. CrTK16.

**Figure 4 genes-13-01676-f004:**
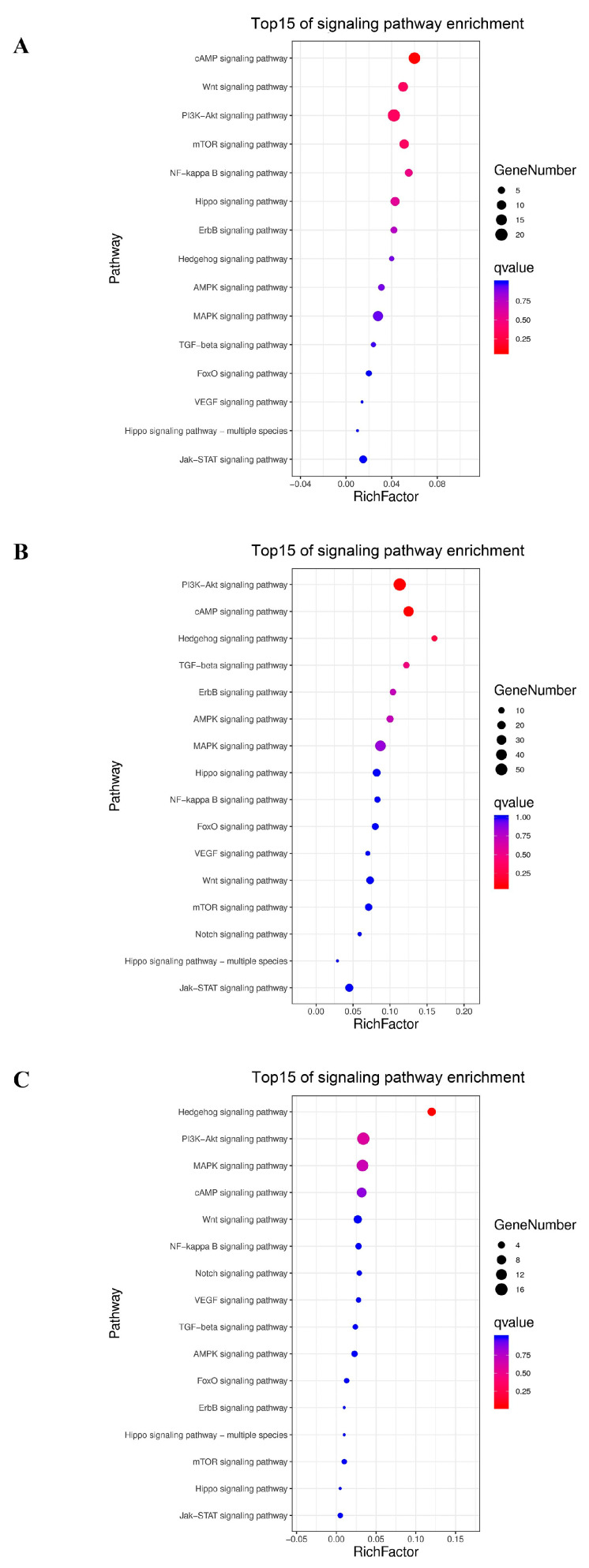
The top 15 pathways of signaling pathway enrichment test in three comparison groups. (**A**) The top 15 of signaling pathway enrichment test for CrTK14 vs. CrTK15. (**B**) The top 15 of signaling pathway enrichment test for CrTK14 vs. CrTK16. (**C**) The top 15 of signaling pathway enrichment test for CrTK15 vs. CrTK16.

**Figure 5 genes-13-01676-f005:**
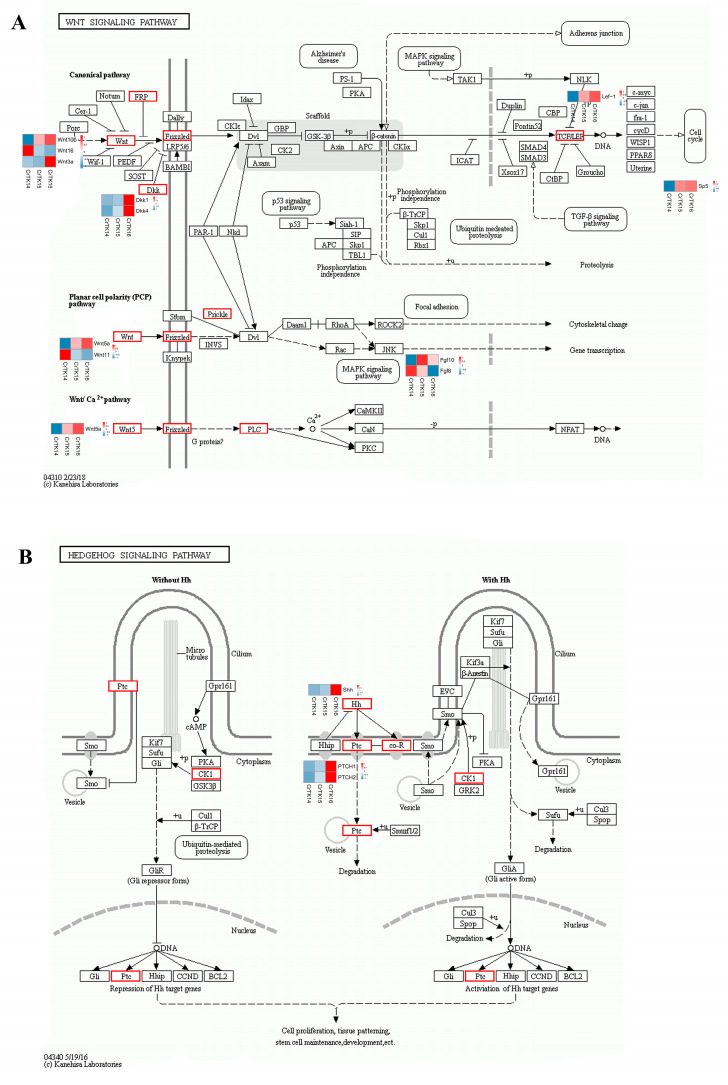
The heat map of the Wnt signaling pathway and the Hedgehog signaling pathway. (**A**) The heat map of the Wnt signaling pathway. (**B**) The heat map of the Hedgehog signaling pathway. The differential expression of genes are marked in red. Heat maps of the genes are shown next to them.

**Figure 6 genes-13-01676-f006:**
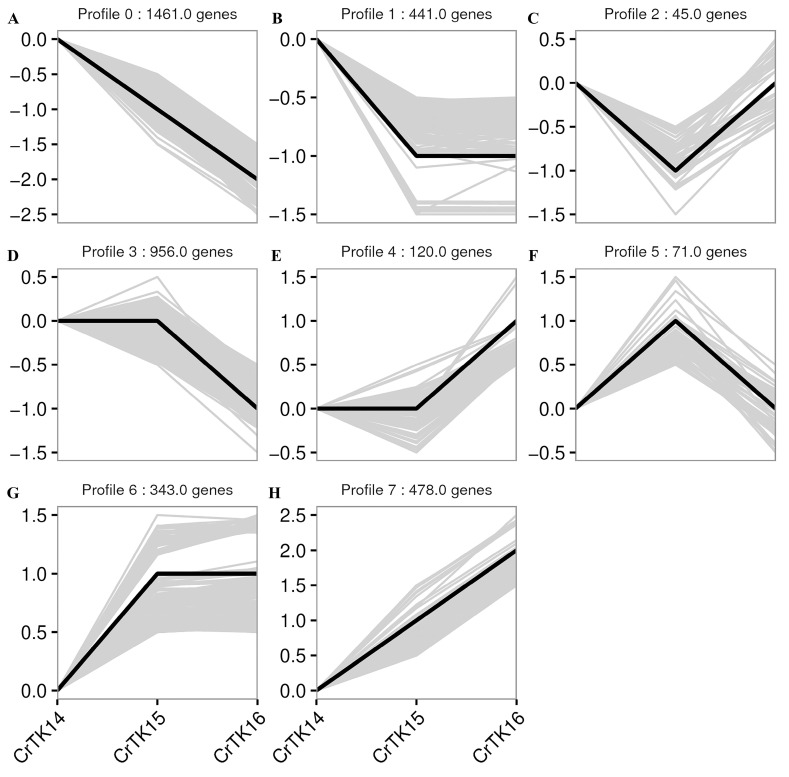
Gene expression profiles of the DEGs. (**A**–**H**) The gene expression Profile 1 to Profile 7 during three stages. The gene data were normalized according to the trend. The ordinate represents the gene expression value. The black curve lines represent the trends, and the gray lines represent each gene in the profile.

**Figure 7 genes-13-01676-f007:**
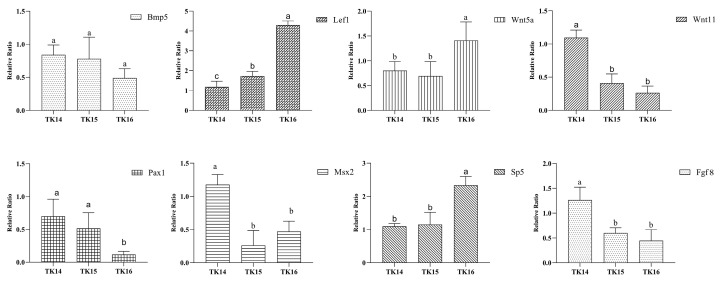
The expression of eight genes in three stages of *M. reevesii* carapacial ridge development using a quantitative real-time reverse transcription-PCR. The means that have different superscript letters are significantly different (*p* < 0.05).

**Figure 8 genes-13-01676-f008:**
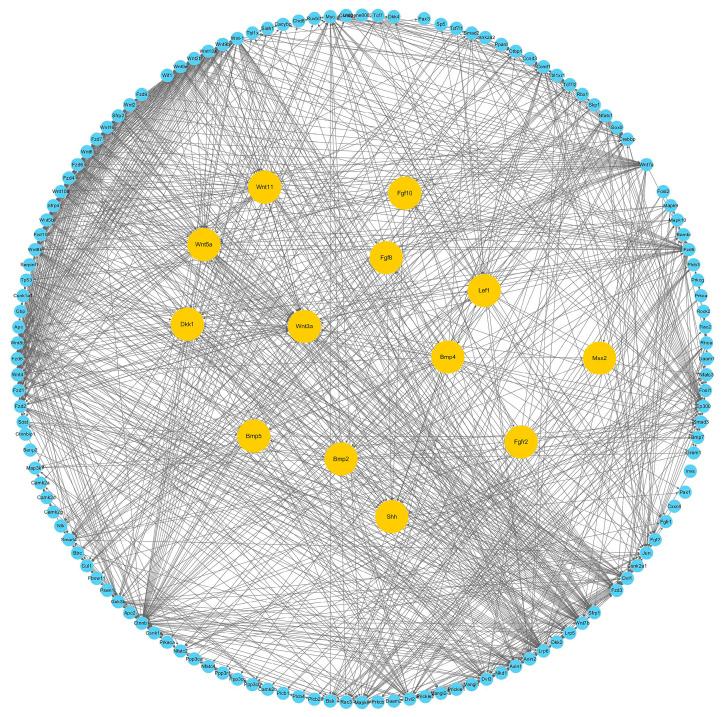
Interaction network of the Wnt signaling pathway. Genes *Bmp4*, *Bmp2*, *Shh*, *Lef1*, *Msx2*, *Bmp5*, *Fgf8*, *Fgf10*, *Wnt5a*, *Wnt11*, *Wnt3a*, *Fgfr2*, and *Fgfr1* are highlighted in yellow, and other genes are highlighted in blue. These relationships scored more than 900.

**Figure 9 genes-13-01676-f009:**
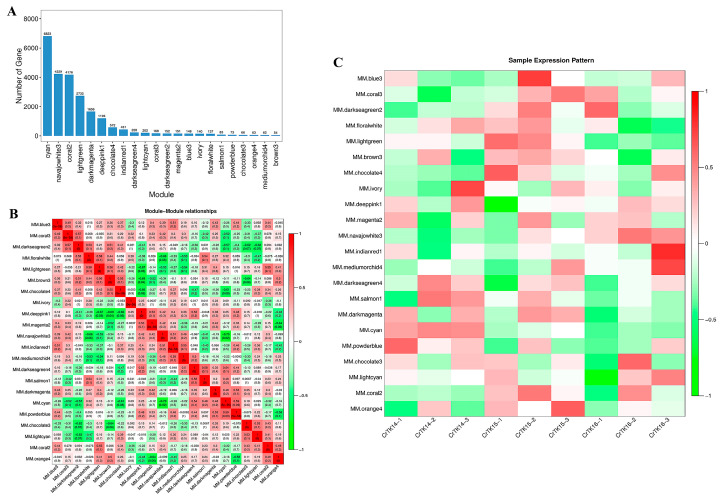
Weighted gene co-expression network analysis (WGCNA) of *M. reevesii*. (**A**) Twenty-two gene modules of *M. reevesii*. The *X*-axis represents the gene module name, and the *Y*-axis represents the number of each module. (**B**) Relationships among the modules. Red represents a positive relationship, and green represents a negative relationship. (**C**) Analysis of the relationship of the gene modules and the CR samples. Each row represents a gene module, and each column represents a sample. Red represents a positive relationship, and green a represents negative relationship.

**Figure 10 genes-13-01676-f010:**
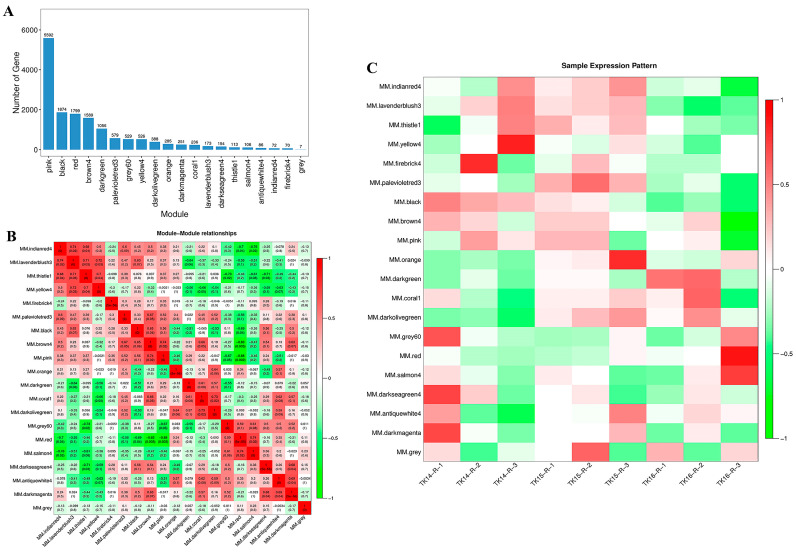
Weighted gene co-expression network analysis (WGCNA) of *P**. s**inensis*. (**A**) Twenty gene modules of *P**. s**inensis*. The *X*-axis represents the gene module name, and the *Y*-axis represents the number of each module. (**B**) Relationships among the modules. Red represents a positive relationship, and green represents a negative relationship. (**C**) Analysis of the relationship of the gene modules and the CR samples. Each row represents a gene module, and each column represents a sample. Red represents a positive relationship, and green represents a negative relationship.

**Figure 11 genes-13-01676-f011:**
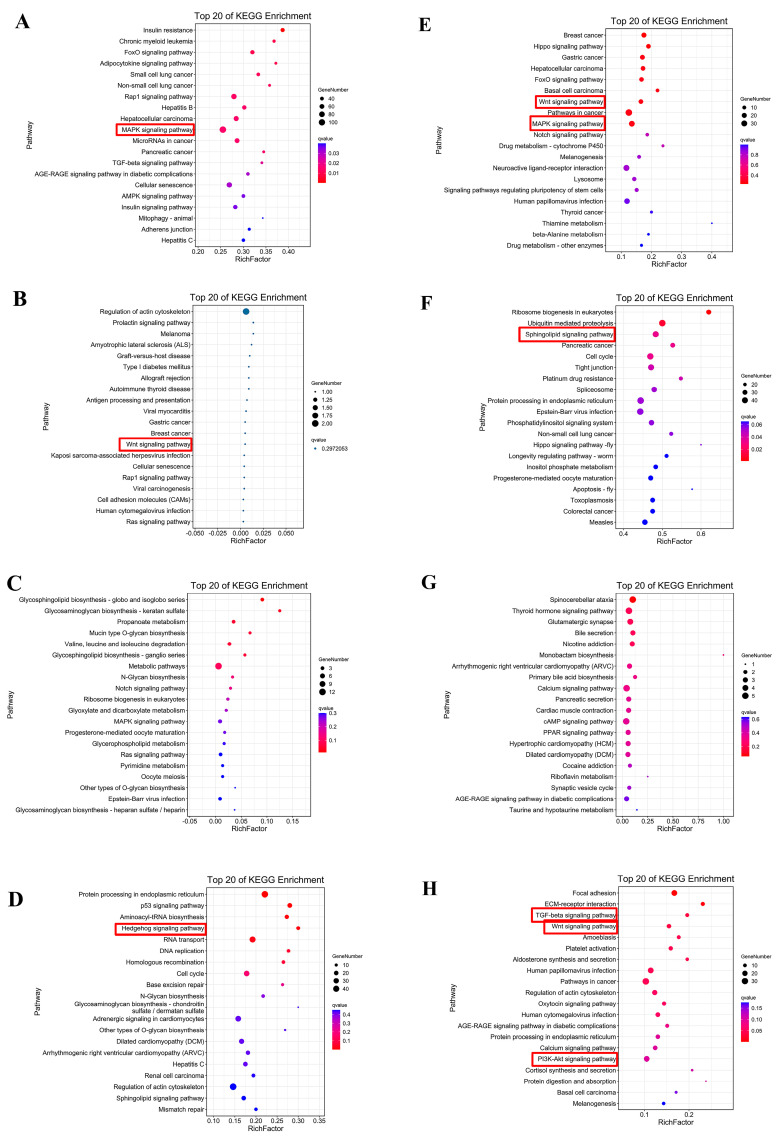
Kyoto Encyclopedia of Genes and Genomes (KEGG) functional enrichment analysis of the key gene modules of *M. reevesii* and *P**. s**inensis*. (**A**) Top 20 results of KEGG enrichment of cyan module. (**B**) Top 20 results of KEGG enrichment of brown 3 module. (**C**) Top 20 results of KEGG enrichment of coral 3 module. (**D**) Top 20 results of KEGG enrichment of navajo white 3 module. (**E**) Top 20 of KEGG enrichment of black module. (**F**) Top 20 results of KEGG enrichment of pink module. (**G**) Top 20 of KEGG enrichment of coral 1 module. (**H**) Top 20 results of KEGG enrichment of dark green module.

**Table 1 genes-13-01676-t001:** Primer sequences that were used in the present study.

Gene	Forward Primer	Reverse Primer
*Fgf8*	CAACGGCAAAGGCAAAGACT	CGGGTGAAAGCCATGTACCA
*Bmp5*	AGTAGAACGCAGCATAGCCC	CCATCAGGATTCTTCCAGGATG
*Msx2*	ATTCAAGAGGCCGGGAGATA	GGGTTCTGGGCTTCCTGTTA
*Wnt5a*	CAGAGCACACTGTTCGGTGA	TCAAAGCAGGATGTGACCCAT
*Wnt11*	GTGGCTGTGGAAAAATGAAGC	CTGATTCAGAGCAAGGACGG
*Sp5*	TCTCCCGCTCTCTGAACACA	GGGTTGAGTGATGGGCATTC
*Pax1*	CGCCTCTTTGAATGCAGTTGT	TTCCAGTCCCGTAGCCAAAC
*Lef1*	GCATCCTCCAGCTCCTGATAT	CTGCCTGAATCCACTCGAGAT
*GAPDH*	CAGAGATGACGACACGCTTAG	CCATTTTCCAGGAGCGTGAT

**Table 2 genes-13-01676-t002:** Raw reads and clean reads of the RNA-sequencing of the CR of *M*. *reevesii*.

Sample	Raw Reads	Clean Reads (%)	Low Quality (%)	Adapter (%)
TotalCrTK14	119,083,328	118,663,122 (99.65%)	321,774 (0.27%)	98,160 (0.09%)
TotalCrTK15	112,965,840	112,517,582 (99.60%)	339,726 (0.30%)	107,652 (0.10%)
TotalCrTK16	119,310,526	118,880,308 (99.64%)	325,846 (0.27%)	102,934 (0.09%)

**Table 3 genes-13-01676-t003:** Key genes of the high relationship gene modules of the CR samples of *M*. *reevesii* and *P*. *sinensis*.

Key Genes in *M. reevesii*		Key Genes in *P. sinensis*	
Cyan	*Wnt5a*, *Wnt11*, *Wnt-1*, *Wnt4*, *Wnt6*, *Wnt10b*, *Wnt16*, *Fgf5*, *Fgf6*, *Fgf20*, *Lef1*, *Mapk8* (JNK);	Black	*Wnt5b*, *Wnt11*, *Wnt3*, *Wnt4i*, *Fgf7*, *Fgf6*, *Fgf19*, *Lef1*;
Dark magenta	*Mapk9* (JNK), *Fgfr1*, *Wnt2b*;	Pink	*JNK*, *Fgfr2*, *Wnt16*;
Coral 3	*Gremlin*1;	Coral1	*Col1a*, *Itga7*;
Navajo white 3	*Shh*, *Ptch1*, *Ptch2*, *Col1a1*, *Col1a2*, *Tgfb2*, *Itga2*, *Itga5*, *Itga7*, *Itga10*, *Itga11*.	Dark green	*Col1a1*, *Col1a2*, *Col24a1*, *Col5a2*, *Itga8*, *Itga11*, *ItgaV*, *Smad6*, *Smad4*, *Tgfb2*, *Bmp6*, *Wnt5a*, *Dkk2*.

## Data Availability

The transcriptome sequencing data of carapacial ridge from *M. reevesii* embryos were submitted to the NCBI sequence Read Archive. (GenBank accession no. SRP310345).

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
