# Peer review of "Gene Regulation during Carapacial Ridge Development of *Mauremys reevesii*: The Development of Carapacial Ridge, Ribs and Scutes"

_genes, 2022, doi:10.3390/genes13091676_

Round 1
Reviewer 1 Report
Kind regards,
Ms. Zinnia Liu
Assistant Editor
In consideration of the paper that I was entrusted to review entitled “Gene regulation during carapacial ridge development of Mauremys reevesii: the development of carapacial ridge, ribs and scutes” by Jiayu Yang , Yingying Xia , Shaohu Li , Tingting Chen , Jilong Zhang , Zhiyuan Weng , Huiwei Zheng , Minxuan Jin , Chuanhe Bao , Shiping Su , Yangyang Liang , Jun Zhang, I tell you that it is a current, detailed, novel and important paper for the readers of the journal and potentially for the experts in the area of study of turtles.
The paper shows us several aspects, the first was to establish an opportunity in the field of embryonic development to show that more studies were needed on the understanding of the metabolic processes in the generation of hard (bone) and soft (skin) shells.
The second aspect was to know the different stages in which the carapacial rigid can be studied and how to establish the collection of the sample in these 3 different stages of development of the shell.
The third thing is that we can use tools of molecular biology, bioinformatics and global databases of both proteins and nucleic acids to compare the results obtained with those previously obtained or to know those missing data and propose novel results at the time of conclusion.
Therefore, in my opinion, even when there is strong and constant evidence on the participation of various genes that turn on and off on what is called canonical and non-canonical Wnt signaling, it is essential to broaden the discussion on the fact of which are the factors that initiate the signaling in these routes to finally obtain a rigid or soft shell, apart from the genes already described in this work. Please extend the discussion of mechanism of stage 16 CR samples of 560 P. sinensis contained Col1a1, Col1a2, and Itga8. Could this mechanism also be inferred for Dermochelys coriacea?
Any idea if any of the pathways described in this paper are affected in those hatchling turtles whose plastron is not fully developed, that therefore the plastron does not "close" completely and the internal organs are seen? Explain briefly.
Linea 189: Get a better resolution of the formula used
Figue 3: Get a better resolution
Line 569: verify the word “reguating”
Line 577: verify the word “to15”
Author Response
Response to Reviewer 1 Comments
Point 1: It is essential to broaden the discussion on the fact of which are the factors that initiate the signaling in these routes to finally obtain a rigid or soft shell, apart from the genes already described in this work.
Response 1: Thank you for your recommendation. Most of factors that initiate the differentiation of hard shell turtle and soft shell turtle appear at stage 15. Different signaling pathways may lead to such differentiation, or different expression sites determine the CR development. For example, Moustakas demonstrated that the continuant and segmental expression of Fgf signal in soft-shell turtle and hard-shell turtle, respectively. Therefore, the analysis need more evidence, we can only make a small assumption for now. The additional explanation has been added in manuscript and highlighted in red. Such as,
Line 524-526: They also found the expression of Shh segmented pattern were lost in CR with inhibition of Bmp and Hedgehog signaling. Besides, this Shh segmented expression were also absent in embryo of P. sinensis [16].
Line 550-552: Experiments disturbed Shh, Bmp and Fgf signaling, leading to the destruction of segmental pattern expression and scutes development [16].
Line 598-599: In addition, the different expression location of key genes expressed in soft shell turtle and hard shell turtle also contribute to this differentiation.
Point 2: Please extend the discussion of mechanism of stage 16 CR samples of 560 P. sinensis contained Col1a1, Col1a2, and Itga8.
Response 2: Thank you for your recommendation. This has been added to lines 586-594 and is highlighted in red.
Point 3: Any idea if any of the pathways described in this paper are affected in those hatchling turtles whose plastron is not fully developed, that therefore the plastron does not "close" completely and the internal organs are seen? Explain briefly.
Response 3: Thank you for your recommendation. In this study, the transcriptome of carapacial ridge was analyzed. It only appears in embryonic stage 14 and stage 16, and may be involved in the development of ribs and scutes, but no clue was found to regulate in plastron closing. Thus, even in the absence of CR, internal organs may remain invisible.
Point 4: Linea 189: Get a better resolution of the formula used
Figue 3: Get a better resolution
Line 569: verify the word “reguating”
Line 577: verify the word “to 15”
Response 4: Thank you for your recommendation. We changed the following points.
- It has been revised in the original manuscript.
- It has been revised in the original manuscript.
- It has been revised in the original manuscript.
- It has been revised in the original manuscript.

Reviewer 2 Report
Gene regulation during carapacial ridge development of Mauremys reevesii: the development of carapacial ridge, ribs and scutes by Yang et al presents the original study on the carapace development of turtles.
It is generally well designed and and it can be improved in below points.
There are some typho mistakes such as double dot in Figure 4 legand and there is no space between profile2 in some cases. These needs to be corrected.
One more thing is, the manuscript is so long and there are many analyses but in the results and discussion section, there is a need to stress on the differences among the stages and different genes. This would be very useful.
The manuscript can be shortened in general.
